# Mucosal Responses to Zika Virus Infection in Cynomolgus Macaques

**DOI:** 10.3390/pathogens11091033

**Published:** 2022-09-12

**Authors:** Neil Berry, Monja Stein, Deborah Ferguson, Claire Ham, Jo Hall, Elaine Giles, Sarah Kempster, Yemisi Adedeji, Neil Almond, Carolina Herrera

**Affiliations:** 1Division of Infectious Disease Diagnostics, National Institute for Biological Standards and Control (NIBSC), Potters Bar EN6 3QC, UK; 2Department of Medicine, Imperial College London, London W2 1PG, UK; 3Division of Analytical and Biological Sciences, NIBSC, Potters Bar EN6 3QC, UK

**Keywords:** Zika virus, mucosal tissue, ex vivo challenge, immune responses, non-human primates

## Abstract

Zika virus (ZIKV) cases continue to be reported, and no vaccine or specific antiviral agent has been approved for the prevention or treatment of infection. Though ZIKV is primarily transmitted by mosquitos, cases of sexual transmission and prolonged viral RNA presence in semen have been reported. In this observational study, we report the mucosal responses to sub-cutaneous and mucosal ZIKV exposure in cynomolgus macaques during acute and late chronic infection. Subcutaneous challenge induced a decrease in the growth factor VEGF in colorectal and cervicovaginal tissues 100 days post-challenge, in contrast to the observed increase in these tissues following vaginal infection. This different pattern was not observed in the uterus, where VEGF was upregulated independently of the challenge route. Vaginal challenge induced a pro-inflammatory profile in all mucosal tissues during late chronic infection. Similar responses were already observed during acute infection in a vaginal tissue explant model of ex vivo challenge. Non-productive and productive infection 100 days post-in vivo vaginal challenge induced distinct proteomic profiles which were characterized by further VEGF increase and IL-10 decrease in non-infected animals. Ex vivo challenge of mucosal explants revealed tissue-specific modulation of cytokine levels during the acute phase of infection. Mucosal cytokine profiles could represent biosignatures of persistent ZIKV infection.

## 1. Introduction

ZIKV was declared a public health emergency by the WHO in 2015/2016 and was linked to the increased incidence of microcephaly in newborns in the Americas [1]. Although first isolated 1947 in the Ugandan Zika forest in a sentinel rhesus monkey [2], ZIKV disease was long associated with mild non-specific syndromes. Phylogenetic analysis groups ZIKV into two lineages: African and Asian [3]. The latter caused smaller outbreaks in Oceania/Pacific and the recent outbreak in the Americas [4].

Vector-borne transmission occurs primarily via Aedes spp. mosquitos, particularly Aedes aegypti. Congenital transmission was confirmed during the 2015/16 outbreak in South America with ZIKV RNA detected in amniotic fluid, placenta and brain tissues of infants born with microcephaly [4,5]. Animal studies in mice and non-human primates (NHPs) confirmed the high efficiency rate of mother-to-fetus transmission [6,7,8,9]. Sexual transmission has been demonstrated in case reports of male-to-female, female-to-male and male-to-male transmissions [4,5,10,11,12,13,14,15]. Furthermore, viral RNA has been detected in male and female reproductive tracts, especially in semen, even after symptoms and viremia had cleared [14,16,17,18,19]. These reports of ZIKV sexual transmission were confirmed in animal studies. ZIKV replication was observed in vaginal tissues following vaginal inoculation of immunocompetent mice [9,20], and Duggal et al. showed male-to-female sexual transmission in mice [21]. High infection rates have been observed in rhesus, cynomolgus macaques and African green monkeys following atraumatic vaginal or rectal challenge [22,23], and persistence of ZIKV RNA after clearance of viremia has been described in NHPs male and female reproductive tracts [24], where the reproductive tracts of both New and Old World NHP species have been demonstrated to be positive for Zika genome and antigen.

Hence, while the majority of transmitted infections will be by mosquito bites, inflammatory responses at mucosal sites where the virus is known to reside may be important in the establishment of persistent infection reservoirs. The localized nature of infection and sampling availability to characterize immune responses at mucosal sites has made studies of this kind difficult to perform; therefore, additional ex vivo approaches may be valuable in this setting.

Mucosal tissue models are increasingly being used for research of other mucosally transmitted pathogens, such as HIV, and are becoming an important tool for pre-clinical evaluation of candidates in the prevention pipeline as well as in early clinical trials [25,26].

Here, we describe the establishment of a tissue explant model to characterize the mucosal responses induced in the colorectal and female reproductive tracts by in vivo ZIKV challenge or by ex vivo challenge of tissues obtained from uninfected animals.

## 2. Results

### 2.1. Late Mucosal Responses to Sub-Cutaneous ZIKV Challenge

To evaluate the mucosal cytokine secretome during late chronic infection, we cultured colorectal and female genital tract (FGT) tissues resected from naïve cynomolgus macaques and ZIKV PRVABC59-infected animals 100 days post-challenge via the sub-cutaneous route, as described previously [27]. A decreased secretion level of the growth factor VEGF was observed in the lower FGT (vaginal and cervical tissues), reaching statistical significance in colorectal tissue (*p* = 0.0294), in contrast to increased levels measured in uterine tissue (Figure 1, Appendix A). Furthermore, there was a greater trend in colorectal explants of a decrease in the levels of inflammatory cytokines (IL-6, IL-1β), chemokines (IL-8, MIP-1β) and the growth factor GM-CSF than in lower and upper FGT tissues. No significant variability was observed between macaques from the same treatment group (Appendix A). With the panel of cytokines used in this study, the top canonical pathways associated with late chronic infection 100 days post-subcutaneous challenge were linked with a non-inflammatory colorectal environment and included the downregulation of pathways associated with IL-17, high mobility group box 1 (HMGB1), triggering receptor expressed on myeloid cells 1 (TREM1), IL-6 and IL-15 signaling, as well as downregulation of crosstalk between dendritic cells and natural killer cells and T helper (Th) cells pathways, among others (Figure 2). A similar pattern was observed in cervical and uterine tissues; however, in vaginal tissue, there was a trend of activation of IL-17- and IL-15-associated pathways, dendritic cell maturation, crosstalk between dendritic cells and natural killer cells, and of Th17 cells.

### 2.2. Infection of NHPs via Intravaginal Challenge

To compare the impact of the challenge route on the mucosal cytokine responses, we set up a vaginal challenge study. A total of six juvenile cynomolgus macaques initially challenged in pairs with one of three different doses of the PRVABC59 challenge stock via the intravaginal route (Figure 3a) were followed for the presence of a systemic infection as assessed by plasma ZIKV RNA levels up to the second challenge. The outcome of infection determined by the presence or absence of detectable viral RNA in the 10^6^, 10^5^ and 10^3^ pfu groups indicated no evidence of plasma RNA at or below the limit of RT-qPCR (50 RNA copies/mL). Subsequently, 35 days after the first challenge, all six macaques were re-challenged with 3.1 × 10^6^ virus. Figure 3b shows transient “blips” of detectable RNA in R7, R10, R14 and R15 at different times in the first week after the second challenge. Only R14 and R15 initially exposed to 10^3^ of virus inoculum exhibited a profile compatible with productive infection. In both cases, however, the ramp-up in viraemia was delayed at 4 and 6 days post-inoculation, respectively. Comparison with serological profiles of serum IgG and IgM (Figure 4) indicate that only R14 and R15 mount a strong seroconversion response, although interestingly, R7, with the blip of serum RNA, displays a low-level profile of IgG production. Overall, compatible molecular and serological data indicate a productive infection in macaques R14 and R15. Prior to the second challenge, however, there was a transient increase in IgM production in R6 and subsequently in R10, which in R10 remained above background but at low levels.

### 2.3. Late Mucosal Responses to Vaginal ZIKV Challenge

The cytokine levels measured in colorectal and FGT tissues from vaginally infected NHPs 100 days post-challenge differed from those observed in animals challenged subcutaneously (Figure 1 and Appendix A). In all mucosal tissues included in this study, a pro-inflammatory environment was observed with significantly increased secretion of chemokine MIP-1β (*p* < 0.0001 for colorectal, vaginal, cervical and uterine tissue), adaptive cytokine IL-2 (*p* = 0.0002 for colorectal, =0.0004 for vaginal, =0.0018 for cervical and <0.0001 for uterine tissue) and growth factor VEGF (*p* = 0.0044 for colorectal, =0.0261 for vaginal, =0.0058 for cervical and =0.0273 for uterine tissue), and significantly decreased levels of anti-inflammatory cytokine IL-10 (*p* = 0.001 for colorectal, =0.0362 for vaginal, <0.0001 for cervical and =0.0344 for uterine tissue). This cytokine profile was linked to the activation of biological pathways including IL-17 and IL-6 signaling, wound healing signaling and HMGB1 and TREM1 signaling, and the activation of Th1 and Th17 cells, among others (Figure 2).

However, in the FGT of animals not productively infected 100 days post-vaginal challenge, VEGF levels were significantly greater than those from infected animals (*p* = 0.0101 for vaginal, =0.0069 for cervical, <0.0001 for uterine tissue), and levels of IL-1β, IL-1ra and IL-10 were lower (reaching statistical significance for IL-10 (*p* = 0.0002) and IL-1ra (*p* = 0.0016) in vaginal tissue and for IL-10 (*p* = 0.0023) and IL-1β (*p* = 0.0003) in uterine tissue) (Figure 1). These changes were linked to lower levels of activation of biological pathways associated with IL-6 and IL-15 signaling and wound healing signaling, compared to the activation levels observed during late chronic infection following vaginal exposure to ZIKV (Figure 2). In the colorectal tract, the trend of reduced secretion of IL-1β, IL-1ra and IL-10 in uninfected animals compared to those infected was in a context of decreased levels of VEGF (*p* = 0.0047), GM-CSF (*p* = 0.0274), IL-6 (*p* = 0.1187) and IL-8 (*p* = 0.2981) (Figure 1). This cytokine profile was linked to a similar modulation of biological pathways as that induced by subcutaneous challenge, with a downregulation of IL-17, IL-6, IL-15, TREM 1 and wound healing signaling, and of Th2 activation, crosstalk between dendritic cells and natural killer cells, and the T cell exhaustion signaling pathway (Figure 2).

### 2.4. Early Responses to ZIKV Challenge in a Mucosal Tissue Model

To determine if mucosal infection could mimic an ex vivo tissue model, we cut explants from colorectal, vaginal, cervical and uterine tissues resected from naïve animals and challenged these tissue explants ex vivo with the same viral stock of ZIKV PRVABC59 used for in vivo studies. Different kinetics of infection were observed in each mucosal tissue when explants were cultured for 21 days post-viral challenge (Figure 5). Significantly higher ZIKV RNA levels were observed in colorectal and uterine explant culture supernatants than in cervicovaginal explant cultures at day 21 post-challenge (when comparing viral RNA levels in colorectal *p* = 0.0052 and uterine *p* < 0.0001 vs. cervicovaginal; Figure 5a). The same pattern was observed when measuring viral titers of culture supernatants during the 21 days of culture (Figure 5b), with higher titers in colorectal (*p* = 0.1778) and in uterine explant supernatants (*p* = 0.0004) than in cervicovaginal culture supernatants.

Viral RNA in situ hybridization (ISH) via RNAscope ISH was conducted to determine the presence of ZIKV envelope (Env) RNA in tissue explants challenged ex vivo. ZIKV Env RNA was detected in the vaginal stratified epithelium and in the colorectal lamina propria (Figure 6; negative controls with un-challenged explants and challenged without RNA probe are shown in Appendix A) at day 1 post-challenge. Furthermore, immunohistochemistry analysis confirmed the presence of replicating ZIKV in the explants with positive staining of NS1 protein at days 3 and 7 post-challenge in both colorectal and FGT explants (Figure 7).

We then assessed the early responses to ex vivo challenge in this tissue explant model 24 h post-exposure to ZIKV PRABC59 (Figure 8, Appendix A). The pro-inflammatory profile measured in vaginal explants following ex vivo challenge was similar to the response observed during chronic infection following vaginal challenge. The significant increases in MIP-1β (*p* = 0.0278), IL-6 (*p* = 0.0087), IL-2 (*p* = 0.0499) and IFN-γ (*p* = 0.0063) and the trends of upregulated secretion of VEGF (*p* = 0.5034) and downregulation of IL-10 levels (*p* = 0.3451) were associated with activation of IL-17 and IL-6 signaling pathways as well as wound healing, HMGB1 and TREM1 signaling, and activation of Th1 and Th17 cells, among others. This pro-inflammatory profile was not observed in the other FGT tissues. Ex vivo challenge induced non-significant changes in cervical explant supernatants with decreased secretion of IL-6, MCP-1 and IL-2 and increased secretion of VEGF, IFN-γ and IL-10. In uterine explants, a non-significant decrease in IL-10 levels and increases in IL-15 and IL-1ra were measured in culture supernatants at day 1 post-challenge.

The early responses to ex vivo ZIKV exposure of colorectal explants were characterized by the significant decreases in IL-6 (*p* = 0.0469) and VEGF (*p* = 0.0376) accompanied by a trend of decreased secretion of IL-8, IL-1β and IL-1ra (Figure 8a). These changes were associated with the downregulation of pathways associated with IL-17, IL-6 and IL-15 signaling, wound healing, HMGB1 and TREM1 signaling, as well as downregulation of crosstalk between dendritic cells and natural killer cells and of dendritic cell maturation, among others (Figure 8b).

## 3. Discussion

Understanding immunological and cellular processes that impact mucosal infection following ZIKV infection and responses generated locally could provide insight into factors that determine infection outcomes via this route related to the virus. Studies which directly address this are difficult to perform and model systems may be useful in deciphering these responses. Here, we employed a combined approach of both in vivo challenge observational studies and explant systems to address this.

While IFN deficiency is required to observe ZIKV-induced disease in mice [28], several ZIKV mouse models have been developed; however, NHPs represent fully immunocompetent hosts and as such are deemed more clinically relevant [27,29,30,31]. In the in vivo intravaginal challenge study, a mixed picture emerged using plasma viral RNA and systemic serological responses as surrogate markers for infection following mucosal exposure to Zika virus at different doses. Following the first exposure to higher doses of 10^6^ or 10^5^ pfu, there is some evidence of low-level, transient antibody production that is detectable in the serum. The lack of detectable virus by sensitive molecular techniques suggests that this may be produced at local sites and therefore only present at low levels in the blood. At the 10^3^ pfu dose, however, there is no evidence of RNA or antibody detected up to the time of the second challenge. After a second challenge at the highest 10^6^ pfu dose, only the macaques initially challenged with the 10^3^ pfu doses showed evidence of systemic virus infection, accompanied by a strong seroconversion for both IgM and IgG and a rapid increase in serum RNA levels. Compared to direct intravenous inoculation [27], in both macaques, this was delayed and staggered in appearance. These data suggest that a low-level, sub-clinical infection may have occurred in at least a proportion of macaques challenged with 10^6^ or 10^5^ pfu of virus, which may have elicited localized responses at the mucosal surfaces which were not readily detected in the blood, but which may have conferred protection at a local level to re-challenge since only the macaques initially challenged with the 10^3^ pfu dose subsequently succumbed to the second 10^6^ pfu challenge. Although, if this was the case, it is unclear why the initial challenges with 10^6^ pfu of virus inoculum were unsuccessful. Exposure to virus which does not result in infection, but which may stimulate protective responses, has been reported for other scenarios where mucosal responses may be involved, e.g., cases of exposed uninfected HIV or SIV [32,33]. Alternative mechanisms which have been described for other Flaviviruses such as dengue virus, including antibody-dependent enhancement (ADE) of the second infection, may also be at play here, and would merit further study.

Hence, to better understand the nature of responses that may be generated at mucosal sites, ex vivo tissue explant models were established, and cytokine profiling was performed. Surprisingly, our results show that during the late chronic phase of infection, 100 days post-viral exposure, the mucosal cytokine secretome profile was tissue-specific and was still determined by the viral challenge route. Tissue-dependent cytokine profiles are expected; however, the opposite modulation of the growth factor VEGF in the colorectum and lower FGT following subcutaneous infection vs. vaginal challenge could be indicative of distinct viral replication fitness and/or distinct changes in the mucosal environment during the acute phase of infection. Greater levels of VEGF were observed in macaques infected subcutaneously than in those challenged vaginally. This increase in VEGF in the upper FGT could be linked to that observed in human amniotic fluid during the mid-trimester of gestation [34]. Furthermore, modulation of IL-17 signaling in colorectal tissue following vaginal challenge could indicate viral dissemination, which has been previously described [35,36] with different degrees of persistence in both compartments. However, subcutaneous exposure induced a downregulation of inflammatory phenotype, which could suggest an immunosuppressive response in cynomolgus macaques. Despite the well-described inflammatory nature of ZIKV infection systemically, neurologically, and in female and male reproductive tracts, limited studies have been performed in the colorectal tract. Inflammation has been shown with epithelial cell lines [37,38], but not in immunocompetent mice following intravenous inoculation [39]. Furthermore, viral kinetics and immune responses have been shown to be isolate-dependent [40,41]. Interestingly, the responses induced within the first 24 h following ex vivo challenge of vaginal tissue explants were parallel to those measured during the late infection stage in vaginal explants obtained from animals challenged vaginally. This suggests, as it has been reported for other mucosally transmitted viruses such as HIV [26], that mucosal responses during the acute phase of infection are elicited in the first 24 h of transmission, and these early responses could be predictive of those measured at a later stage during chronic infection. In this regard, the cytokine profile observed at 100 days post-challenge would not be anticipated to return to baseline levels pre-challenge considering the chronic nature of ZIKV infection and the possibility of residual low levels of virus sequestered at distal sites. Further in vivo challenge studies will be necessary to compare the mucosal cytokine responses elicited following intrarectal challenge with those measured in the ex vivo explant challenge model. Moreover, our previous NHP studies of subcutaneous ZIKV transmission highlighted the chronic nature of the infection in these hosts, including ZIKV RNA detected in brain and male testes after clearance of the acute infection, as determined by plasma viraemia. Hence, it will be interesting to determine in which sites residual viral levels are responsible for chronic infection following mucosal challenge.

In our in vivo vaginal challenge study, four animals did not become productively infected. Analysis of the mucosal cytokine profiles in both colorectum and FGT revealed differences when compared to the profiles for the two infected animals. Hence, further studies will be necessary to determine if mucosal cytokine levels could constitute a correlate of protection in NHP and humans. Systemic and mucosal protective cellular and humoral immune responses have been described in mice against vaginal ZIKV infection [42]. Modulation of type I IFN responses are known to be directly linked to ZIKV replication capacity [28,42]; however, our cynomolgus cytokine panel did not include any type I IFNs but only IFN-γ, which was not linked to a profile of resistance to ZIKV infection following intravaginal challenge. Studies in mice models have shown that passive transfer of ZIKV-specific IgG was sufficient to prevent intravaginal infection [42]; hence, future studies will need to assess the extent of B cell responses in mucosal compartments capable of preventing sexual transmission.

Initial RNAscope ISH and immunohistochemistry analysis confirmed the presence of ZIKV in tissue explants following ex vivo challenge. However, further studies will be necessary to assess the type of cells infected within the tissue explants.

Numerous ex vivo models of ZIKV infection have been described using placenta or brain tissue [43,44,45,46,47,48,49,50,51,52,53,54], but there is limited literature describing infection of other mucosal tissues [55] and the impact of viral exposure on the mucosal environment. Our ex vivo mucosal tissue explant models of ZIKV challenge for the colorectum and the FGT were adapted from the models used for HIV [56,57,58,59]. The HIV explant models, which include epithelium and lamina propria, have been shown to be an important tool for basic research [26,60,61,62,63,64] and pre-clinical screening of PrEP regimens [65], and are increasingly being used in clinical trials for pharmacological assessment of anti-retrovirals [66,67,68,69,70,71,72,73,74,75,76,77,78,79,80,81]. It has been shown that consistent results can be obtained among different laboratories through protocol standardization [82]. Furthermore, in vivo HIV replication fitness can be mimicked in mucosal tissue explants [25,83,84], and following ex vivo challenge, HIV virions have been shown in different mucosal tissue explant models to penetrate the lamina propria with similar kinetics to those observed in vivo [26,85,86]. These models have limitations, including (i) progressive loss of architecture despite the maintenance of CD4:CD8 T cell ratios and sufficient viability to sustain HIV replication for more than 10 days [87], (ii) paucity of data regarding the preservation of immune competence [57] and (iii) limited ability to demonstrate sterilizing protection when evaluating anti-retrovirals.

Results obtained with ex vivo culture of NHP tissues will need to be confirmed in parallel models using human tissue to define the potential effect of mucosal differences between species. Among other factors, microbial communities have been shown to differ between both species [85], vaginal pH is higher in macaques than in humans [86] and humoral immunobiology differs with the diversity of immunoglobulins and antibody receptors between humans and NHPs [88].

The main limitations of this study are the number of animals included and the limited panel of cynomolgus cytokines evaluated. Despite the use of outbred cynomolgus macaques and the small number of animals per group, no significant inter-individual variability was observed among the cytokines measured. No imaging of cervical tissue was performed due to the required tissue size for RNAscope ISH or immunohistochemistry. Future histological analysis will be interesting to determine the impact of changes in VEGF levels on the tissue architecture, or the potential immune cell infiltration in tissue following in vivo challenge, among other aspects. A comparison between mucosal and systemic responses was not performed assessing cytokine profiles, IgM/G/A levels and B cell phenotypes in both compartments. Our study was not set up to detect neither the first cells targeted by the virus, nor the number of cells productively infected. Future studies will also be necessary to identify the cellular populations responsible for the early responses and their modulation during each phase of infection.

This observational study supports the use of ex vivo mucosal challenge models as surrogates of in vivo transmission to potentially define mucosal correlates of risk or protection against ZIKV infection, which could inform the design of new prophylactics and therapeutics. Furthermore, this pre-clinical ex vivo challenge model would increase the screening capacity of new drug candidates while reducing the number of animals used.

## 4. Materials and Methods

### 4.1. Ethical Statement

All animal procedures were performed in strict accordance with U.K. Home Office guidelines, under license 70/8953 granted by the Secretary of State for the Home Office, which approved the work described. Regular modifications to the housing area and environmental enrichment of all study NHPs were made by husbandry staff. Animal rooms were subject to 12 h day/night cycles of lighting. Animals were acclimatized to their environment and deemed healthy by the named veterinary surgeon prior to inclusion in the study. All surgical procedures were performed under anesthesia with recovery.

### 4.2. Virus and In Vivo Vaginal Challenge

PRVABC59 ZIKV strain (GenBank: KU501215) used in all studies was obtained from National Collection of Pathogenic Viruses, Public Health England, Porton Down, Salisbury, UK [3]. Virus stocks were prepared by propagation and titration on Vero cells (6.32 × 10^6^ plaque-forming units (pfu)/mL). No onward culture of virus or adaptation on macaque cells prior to experiments was performed.

Pairs of female Mauritian-derived cynomolgus macaques were atraumatically inoculated vaginally with either 3.1 × 10^6^ (R6, R7), 3.1 × 10^5^ (R8, R10) or 3.1 × 10^3^ pfu (R14, R15) PRVABC59 virus inoculum, and a second challenge after 35 days with all 6 macaques inoculated with 10^6^ pfu of the same PRVABC59 stock. Sequential blood samples at the time points shown in Figure 3a were collected from macaques into EDTA (vacuette tubes, Greiner bio-one), centrifuged to obtain plasma, aliquoted and frozen immediately at −80 °C. Serum was isolated by centrifugation from blood collected without anti-coagulant and allowed to clot for 4–16 hrs.

### 4.3. Serology

Anti-ZIKV serology was assessed by measuring binding IgG and IgM levels. Total anti-ZIKV IgG levels were determined by ELISA (D-23560, Euroimmun Ltd.) according to the manufacturers’ instructions. Briefly, serum was diluted 1/100 with diluent before incubation on strips coated with NS-1 antigen for 1 h at 37 °C. Bound antibody was detected by the conjugated secondary antibody and substrate addition. Relative units (RU) were calculated from 450 nm absorbance with the subtraction of Abs 630 nm using kit calibrants 1–3 to generate a standard curve. Anti-Zika IgM levels determined by ELISA (Euroimmun Ltd.) were expressed as a ratio.

### 4.4. Cell, Tissue Explant, and Virus Culture Conditions

All cell and tissue explant cultures were maintained at 37 °C in an atmosphere containing 5% CO_2_. Vero cells (ATCC—CCL-81) were kindly donated by Prof. Peter O’Hare. Cells were grown in Dulbecco’s Minimal Essential Medium (DMEM) (Sigma-Aldrich, Inc., St. Louis, MO, USA) containing 10% fetal calf serum (FCS), 2 mM L-glutamine and antibiotics (100 U of penicillin/mL, 100 µg of streptomycin/mL). The cells were tested for mycoplasma contamination and confirmed to be mycoplasma-free.

Mucosal tissue specimens were transported to the laboratory at 4 °C and cut upon arrival into 2–3 mm^3^ explants comprising epithelial and stromal layers for FGT tissue or epithelium and muscularis mucosae for colorectal tissue [83,89]. The tissue explants were maintained with DMEM containing 10% fetal calf serum, 2 mM L-glutamine and antibiotics (100 U of penicillin/mL, 100 µg of streptomycin/mL, and 80 µg of gentamicin/mL).

### 4.5. Infectivity Assay

Tissue explants were challenged with ZIKV 10^5^ pfu/mL for 2 h in a non-polarized system and then washed 4 times with PBS to remove unbound virus. FGT explants were then transferred to fresh 96-well plates, and colorectal explants were deposited onto gel foam rafts (Welbeck Pharmaceuticals, London, UK) in 24-well plates in the absence of virus. Explants were cultured for 21 days with approximately 50% of the supernatants harvested at days 3, 7, 11, 15 and 21, and replaced with fresh media. Infectivity was assessed by plaque assay and quantification of viral RNA content in culture supernatants.

### 4.6. Plaque Assay

Vero cells were seeded at 2.3 × 10^5^ in 12-well plates to reach 90–100% confluency the next day. Cell were washed twice with PBS before adding 200 μL of serially diluted ZIKV stock or tissue explant culture supernatant in DMEM without serum. Plates were incubated at 37 °C for 2 h with gentle rocking every 10–20 min to evenly spread the virus. The supernatant was discarded, and 1.5 mL of a 1:1 mix of pre-warmed DMEM 2% and 2% carboxymethyl cellulose (CMC; Sigma-Aldrich, Inc., St. Louis, MO, USA) solution was added to each well. After 5 days of culture, the DMEM/CMC mixture was discarded and replaced with 1 mL of the fixing/staining solution (0.25% crystal violet, 1.85% formaldehyde, 35 mM Tris, 0.5% CaCl_2_ in ddH_2_O) and incubated for 30 min at room temperature before washing the wells with tap water. Plaques were counted and the titer was calculated as plaque-forming units (pfu) per mL.

### 4.7. Quantification of Viral RNA by RT-qPCR

RNA was extracted from EDTA-treated plasma and culture supernatants with the QIAmp viral RNA mini kit (Qiagen, Hilden, Germany) and from tissue explants with the RNeasy Plus Universal Mini Kit (Qiagen, Hilden, Germany), following the manufacturer’s instructions. Tissue explants were homogenized using QIAzol lysis reagent in 2 mL lysing matrix A tubes (MP Biomedicals, Santa Ana, CA, USA) with a Bio 101 FastPrep FP120 Cell disruption system (ThermoSavant, Waltham, MA, USA). Homogenates were spun down to remove the lysing matrix A prior to RNA purification. RNA was quantified by nanodrop technology.

ZIKV RNA was quantified by one-step RT-qPCR with ZIKV-specific primer sequences using forward (ZIKV 1086) (CCGCTGCCCAACACAAG) and reverse (ZIKV 1162c) (CCACTAACGTTCTTTTGCAGACAT) PAGE-purified primers (Eurofins) [90]. Probe sequence (ZIKV 1107-FAM) (AGCCTACCTTGACAAGCAGTCAGACACTCAA) was quenched with 3′ Iowa Black to further reduce background fluorescence using a second internal ZEN quencher (Integrated DNA Technologies, Inc., Coralville, IA, USA). ZIKV RNA was assayed in duplicate using the RNA Ultrasense one-step qRT-PCR system (Thermo Fisher Scientific, Waltham, MA, USA), adopting the cycling conditions of a 15 min reverse transcription step at 50 °C, followed by a 2 min step of denaturation at 95 °C and 45 cycles of amplification consisting of 95 °C for 15 s and 60 °C for 30 s, as described previously [91]. Copy number values were determined against the ZIKVPRVABC59 stock grown in Vero cells. The amplification profile was as followed using the Stratagene1 Mx3000P qRT-PCR thermo cycler (Thermo Fisher Scientific, Waltham, MA, USA).

### 4.8. RNAscope In Situ Hybridization (ISH) and Immunohistochemistry

Tissue blocks were cut for microscopy analysis. Following ex vivo challenge or not, explants were harvested at different time points and fixed in a 10% formalin solution (Sigma-Aldrich, St. Louis, MO, USA) for 1 to 3 days at room temperature prior to analysis. RNAscope and immunohistochemistry were performed as previously described [27]. Microscopy analysis was performed with a Leica Bond-Max system (Leica, Wetzlar, Germany) and image analysis with Pannoramic Viewer software (3DHistec, Budapest, Hungary).

For RNAscope ISH, fixed tissue blocks were embedded in paraffin wax (VWR) using previously reported procedures [92]. Briefly, 4 µm thick sections were mounted on poly-L lysine-coated slides (VWR) and, prior to treatment, de-waxed in xylene and re-hydrated via graded ethanol:water solutions. In situ ZIKV RNA detection was performed using the RNAscope 2.5 HD manual DAB detection system (Advanced Cell Diagnostics, Newark, CA, USA) and a combination of two ZIKV-specific probes (463781 and 464531) in accordance with the manufacturer’s instructions. Negative (DaPB 310043) and positive (Hs-PPIB 313901) control probes were used to assess technique efficiency. Sections were stained with hematoxylin for the detection of cellular nuclei.

### 4.9. Multiplex Cytokine Analysis

The levels of 23 cytokines (IL-1β, IL-1Ra, IL-2, IL-4, IL-5, IL-6, IL-8, IL-10, IL-12/23(p40), IL-13, IL-15, IL-17, Il-18, sCD40L, G-CSF, GM-CSF, IFN-γ, MCP-1, MIP-1α, MIP-1β, TGF-α, TNF-α and VEGF) in tissue supernatants were quantified after 24 h of culture by an NHP-specific multiplex bead immunoassay (Merck Millipore, Burlington, MA, USA) using a Luminex 200 System (Bio-Rad, Hercules, CA, USA).

### 4.10. Statistical Analysis

Cytokine/chemokine, viral load and antibody concentrations were calculated from sigmoid curve fits (Prism v. 9.2.0, GraphPad, San Diego, CA, USA). All data presented fulfill the criterion of R^2^ > 0.7. The statistical significance of differences between baseline and post-vaccination samples were determined using a non-parametric Kruskal–Wallis test with no correction for multiple comparisons. The responses to vaccination were considered significant when *p* < 0.05.

Heat maps were completed using Prism with protein levels normalized to the matched explant controls instead of the mean of the control treatments to improve the correction for explant effects and log transformed (base 2). Differentially abundant proteins were analyzed using Ingenuity Pathway Analysis software to determine the biological processes affected by vaccination. The pathways with a minimum of at least two associated analytes and *p* < 0.05 were considered to be enriched.

Interindividual variability was calculated using a Spearman correlation test. *p* values were determined with a 95% confidence interval using a two-tailed unpaired Student’s *t* test, and a *p* value < 0.05 was considered statistically significant.

## Figures and Tables

**Figure 1 pathogens-11-01033-f001:**
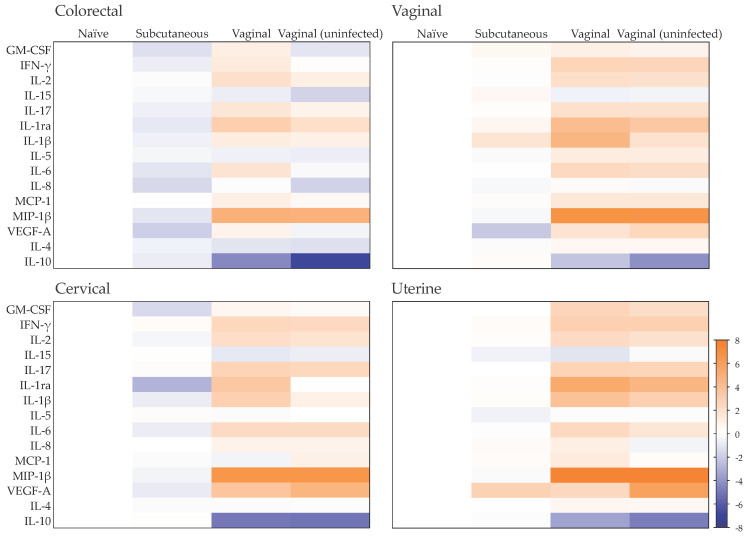
Effect of ZIKV challenge route on the mucosal cytokine profile. Heat map representing cytokines that are upregulated (orange) or downregulated (blue) in culture supernatants from colorectal, vaginal, cervical or uterine tissues obtained 100 days post-challenge from cynomolgus macaques infected or not following subcutaneous or vaginal challenge and from naïve, non-challenged animals. Differences are shown in Log2 from independent experiments performed at least in duplicate with two animals per group.

**Figure 2 pathogens-11-01033-f002:**
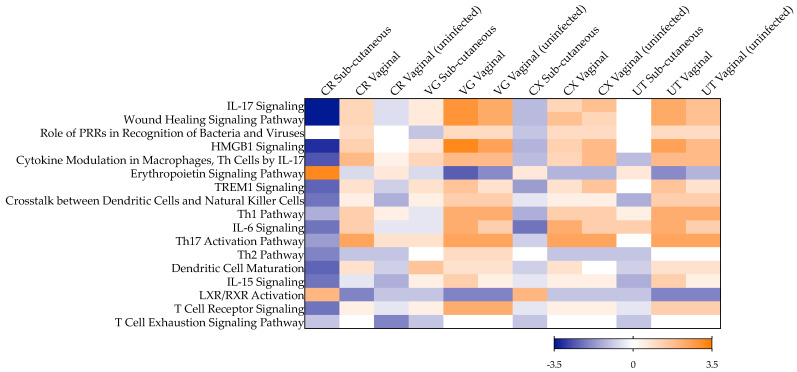
Canonical pathways significantly enriched in mucosal tissues post-ZIKV challenge. Heat map representing the activation z-scores determined by directionality (overabundance in orange and underabundance in blue) and the number of cytokines measured in colorectal (CR), vaginal (VG), cervical (CX) or uterine (UT) tissue culture supernatants. Tissues were obtained 100 days post-challenge from cynomolgus macaques infected or not following subcutaneous or vaginal challenge and compared with tissues from naïve, non-challenged animals. The pathways included had a minimum of two analytes associated and a *p* < 0.05 (right-tailed Fisher exact test).

**Figure 3 pathogens-11-01033-f003:**
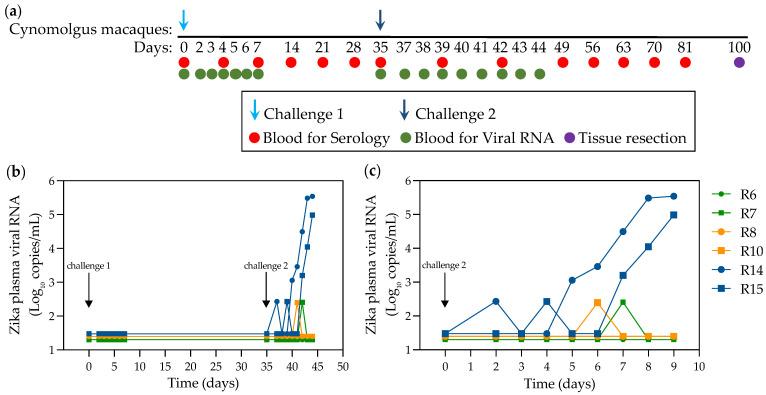
Post-challenge ZIKV RNA level following exposure via the intravaginal route. (**a**) Vaginal challenge and sampling schedule. (**b**) Viraemia profiles across the entire time-course following the first challenge with pairs of cynomolgus macaques initially challenged with 3.1 × 10^6^ (R6, R7), 3.1 × 10^5^ (R8, R10) or 3.1 × 10^3^ pfu (R14, R15) ZIKV PRVABC59 virus inoculum, and a second challenge after 35 days with all 6 macaques challenged with 10^6^ pfu of the same ZIKV PRVABC59 stock. (**c**) Detailed kinetic profiles following the second challenge. Lower limit of detection for the assay is 50 RNA copies/mL.

**Figure 4 pathogens-11-01033-f004:**
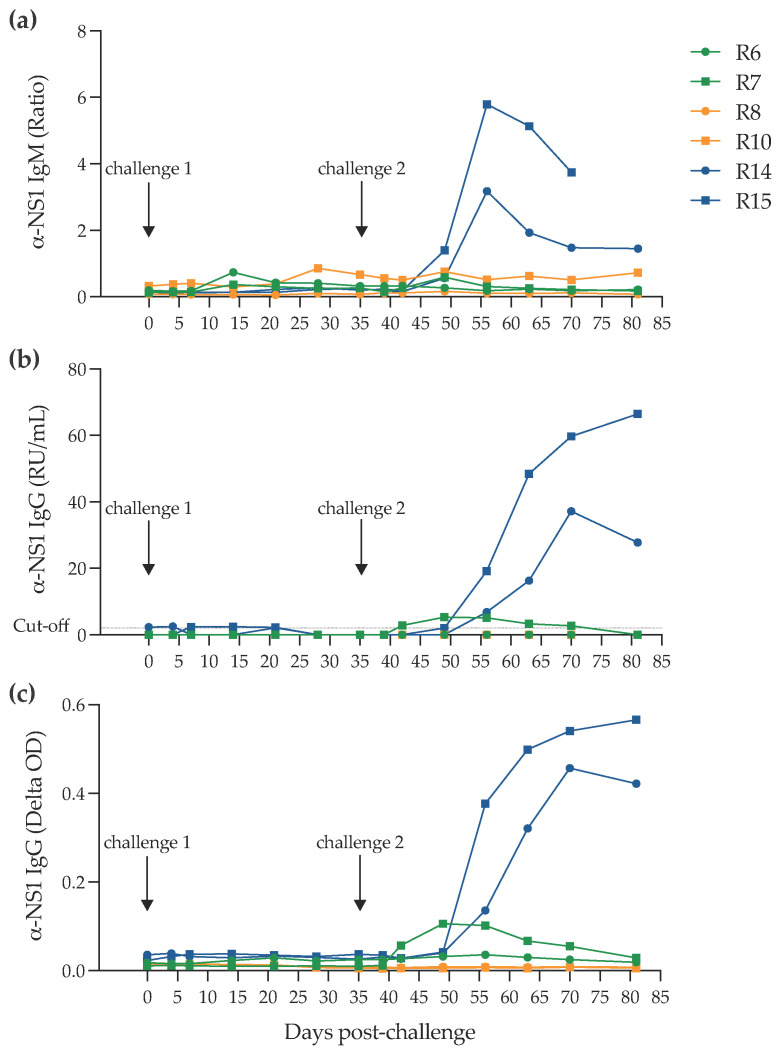
Zika anti-NS1 titers. Zika virus-specific (**a**) IgM levels, (**b**) IgG levels expressed as relative units per mL (RU/mL) and (**c**) IgG levels for the 81 days post-challenge time-course showing bleeding frequencies from the start of the study (day 0).

**Figure 5 pathogens-11-01033-f005:**
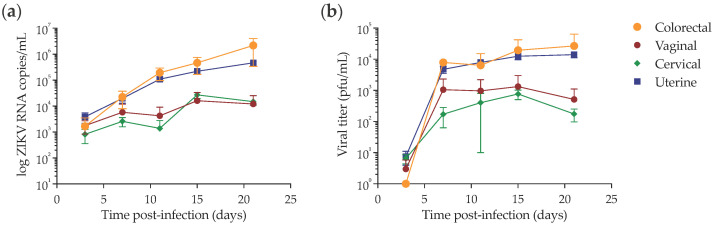
ZIKV replication in mucosal tissue explants following ex vivo challenge. Tissue explants were cultured for 21 days. Culture supernatants were harvested at days 3, 7, 11, 15 and 21. (**a**) ZIKA RNA levels and (**b**) viral titers were measure in the harvested culture supernatants. Data are mean (±SD) of experiments performed at least in duplicate with tissues resected from two animals.

**Figure 6 pathogens-11-01033-f006:**
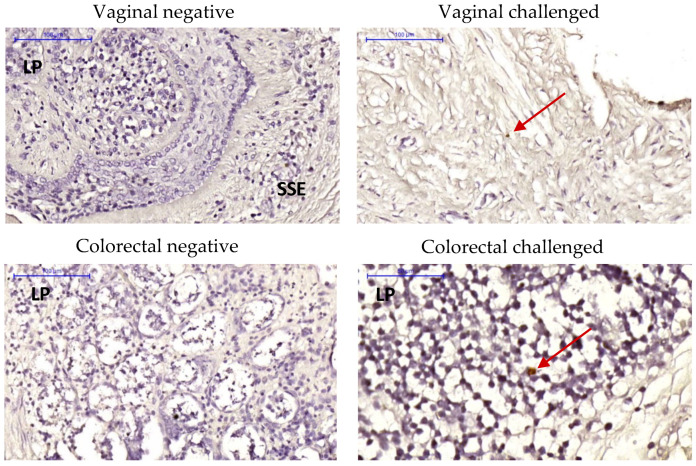
RNAscope in situ hybridization analysis. Vaginal and colorectal explants from naïve animals were cultured for 1 day either unchallenged (negative) or exposed to 10^5^ pfu of ZIKV PRABC59 for 2 h (challenged). Representative infection foci are signaled with red arrows. Nuclei were stained with hematoxylin (blue). Blue bar = 200 μm. SSE: stratified squamous epithelium; LP: lamina propria.

**Figure 7 pathogens-11-01033-f007:**
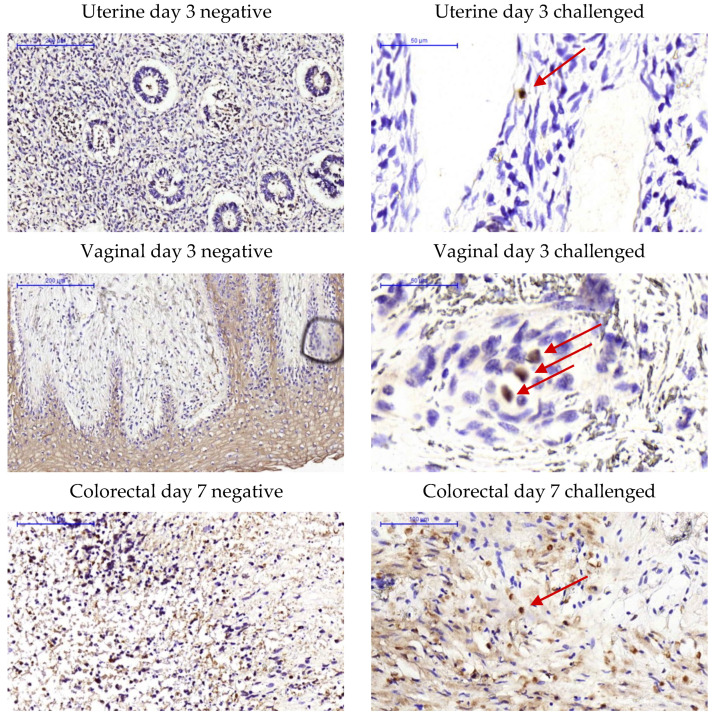
Immunohistochemistry analysis of ex vivo mucosal challenge. Uterine, vaginal and colorectal tissue explants from naïve animals were cultured either unchallenged (negative) or exposed to 10^5^ pfu of ZIKV PRABC59 for 2 h (challenged). Samples were stained with anti-NS-1 antibody. Areas of specific staining are indicated by brown foci. Representative infection foci are signaled with red arrows. Nuclei were stained with hematoxylin (blue).

**Figure 8 pathogens-11-01033-f008:**
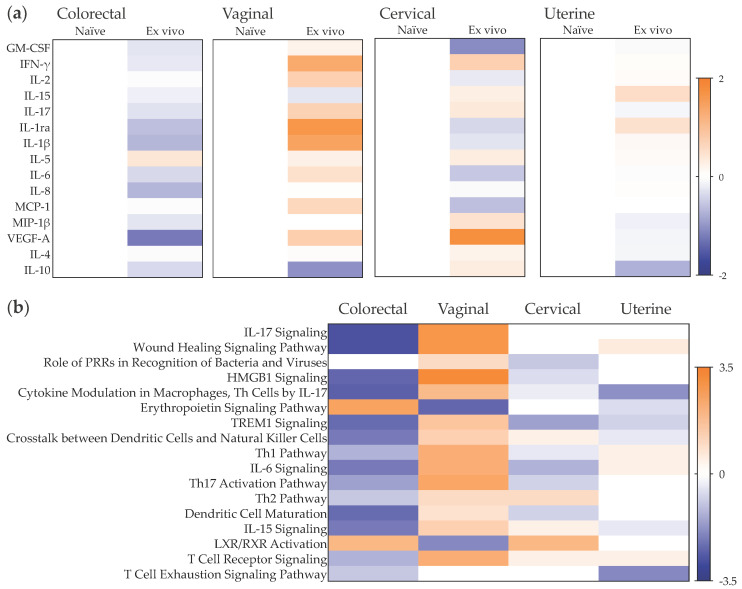
Modulation of mucosal cytokine profiles induced by ex vivo exposure to ZIKV PRABC59. (**a**) Heat map representing cytokines that are upregulated (orange) or downregulated (blue) from cynomolgus macaque colorectal, vaginal, cervical or uterine explant culture supernatants harvested 24 h post-exposure to ZIKV or not. Differences are shown in Log2 from independent experiments performed at least in duplicate with two animals per group. (**b**) Heat map representing the canonical pathway activation z-scores determined by directionality (overabundance in orange and underabundance in blue) and the number of cytokines measured in explant culture supernatants. The pathways included had a minimum of two analytes associated and a *p* < 0.05 (right-tailed Fisher exact test).

## Data Availability

Data available upon request.

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
