# Peer review of "Mucosal Responses to Zika Virus Infection in Cynomolgus Macaques"

_pathogens, 2022, doi:10.3390/pathogens11091033_

Round 1

Reviewer 1 Report

Berry et al. have conducted an interesting study into sexual transmission of ZIKV in a relevant model to humans utilizing macaques. Mucosal infection (opposed to systemic infection) demonstrated significant differences in pathological outcomes. Of note, VEGF expression was increased following vaginal challenge. 

Concerns:

1. Understandably, working with NHP is expensive and difficult (especially during a pandemic), but a sample size of 2 for conditions is limiting the interpretations that can be concluded from the study as the authors acknowledge in the second last paragraph of discussion. This should be presented as an observational study, not as a definitive study. 

2. Whilst there were changes in VEGF were evident (a signalling protein generally associated with endothelial growth, and to some extent the immune system), histopathology of the chronic tissues in situ of vaginal mock and infected animals prior to re-exposure ex vivo should be conducted to determine tissue architecture and remodelling. Perhaps this was performed but doesn’t clearly come across in the text.

3.Th17 signalling in the colon following vaginal challenge is of interest suggesting infection of the colon and polarization of the Th1/Th17 axis in a generally immunosuppressive tissues. Conversely, sub-cutaneous challenge downregulated an inflammatory phenotype in the colon suggesting an immunosuppressive role via the common transmission route of mosquitos. 

4. Vaginal pH of macaques (and specifically cynomolgus macaques) is relatively neutral (5.5-7.5, mean 6.5) which is considerably different to human female pH (4.5). Given this difference, it might be worth addressing this in the discussion.

5. In Figure 1, vaginally uninfected colorectal, vaginal, cervical and uterine tissues all have elevated cytokine expression in correlation to naïve animals which is curious. Given the samples were taken 100 days after exposure to no infection one would expect background levels to be equivalent to naïve animals. The authors should address on this.

5. There should be further discussion on the apparent immunosuppressive role of subcutaneous infection relative to uninfected controls (see Fig 2 – CR Sub-cutaneous vs CR Vaginal (uninfected); CX sub-cutaneous v CX Vaginal (uninfected); UT sub-cutaneous vs UT Vaginal (uninfected).

6. Figure 7 suggests that the unchallenged ex vivo experiment do not expectedly harbour chronic ZIKV infection, but chronic infection and ex vivo rechallenge are re-infected. This might just need clarification in the figure legend as whether the explant tissues all came from chronically infected animals and were rechallenged ex vivo, or if negative controls are true negative controls from uninfected animals that were mock infected ex vivo, or simply just cultured for 2hrs.

Given the chronic nature of the ZIKV infection of the mucosal surfaces and the subsequent investigation of the antigen-specific cytokine and immunoglobulin responses to infection, validation that a chronic infection is established as observed in other studies of immune-privileged sites such as observed in male testes. There should be an investigation into where the depot of the chronic infection is residing (if it still exists) to confirm chronic infection or just chronic immunopathology from previously infected animals, and more importantly are the animals still shedding virus at this late timepoint.

7. Only serology was performed to investigate antibody response, not mucosal luminal secretions. Vaginal and colonic samples should be able to determine this. Investigation of IgA should be undertaken in addition to IgM and IgG as it is a crucial and often dominant arm of the immunoglobulin response in the mucosa, particularly the colonic fluid (331x higher than IgG), uterine cervix (1.7x higher than IgG). In this regard, expression of polymeric immunoglobulin receptor (pIgR) in chronically vaginally-challenged tissues could be performed via immunohistochemistry/qRTPCR as this will have a direct influence on secretory IgA (and IgM) in the luminal secretion of these antibodies in the tissues, and thus the protection by steric blocking by antibodies from ZIKV re-infection. It would also be beneficial to quantify tissue plasma B cells as IgA is mostly produced locally in these tissues.

Reviewer 2 Report

This study by Berry et al characterized cytokine and antibody responses following ZIKV challenge in cynomolgus macaques, using different mucosal routes (colorectal, vaginal, cervical and uterine). They also studied viremia (viral RNA) and serology (anti-ZIKV antibodies) in blood following vaginal challenge. Finally, they tested explants from colorectal, vaginal, cervical and uterine tissues for viral replication and cytokine responses.

I appreciate the authors testing various mucosal routes in NHPs, given the importance of mucosal infection and immunity, a topic important for Zika virus (which is not just transmitted by mosquitoes) and other viral pathogens. Furthermore, the paper suggests mucosal explants support ZIKV replication, which may allow an ex vivo model for further studies.

My main criticism of the study is that there does not appear to be clear or interesting insights taught by the manuscript, as each experiment utilizes a few rather cursory assays/readouts. This is not uncommon for “characterization” studies of disease models, but is an area the authors should consider improving in to boost the importance and translational utility of their findings.

Major comments:

·      The authors need to find some way to present inter-subject variability in their existing figures, which is important for outbred NHPs and has implications for humans/patients. For example, Fig. 1 heat map refers to 2 animals per group. Do the 2 animals have similar magnitude of change in each cytokine? Do the 2 animals in a group occasionally have opposing trends for a cytokine?

·      The authors should also present the statistics for all comparisons they performed in supplementary tables. It is difficult to interpret the cytokine trends highlighted in the text without understanding which comparisons were performed and how their p values/magnitude of change compare to each other.

·      The intravaginal challenge experiment (Fig. 3 and 4) is difficult to interpret, and it raises more questions than it answers. Strangely, 3E6 pfu in a first challenge does not give clear viral RNA in blood, but a similar dose in a second challenge does if the animals received 3E3 pfu in a first challenge, but not 3E5 or 3E6 pfu. Apart from the hypotheses/discussion the authors put forth, could there have been antibody-dependent enhancement (ADE) of the second infection, which is well-described for another flavivirus, dengue virus? Overall, I find this experiment to weaken the paper, and wonder if this surprised finding can be investigated further to lend some novel insights.

Minor comments:

·      Am surprised the authors did not perform immunohistochemical analyses of immune cell infiltration or tissue pathology, which would have illuminated local tissue biology. Any saved fixed tissues could easily undergo such analysis.

·      How did the authors select the time point of 100 days post-challenge?

Round 2

Reviewer 1 Report

The authors have adequately addressed my concerns/comments. Further research is required in this space, and this group is adequately experienced to pursue them.